# Epigenetic Mechanisms of HIV-1 Persistence

**DOI:** 10.3390/vaccines9050514

**Published:** 2021-05-17

**Authors:** Roxane Verdikt, Olivier Hernalsteens, Carine Van Lint

**Affiliations:** Service of Molecular Virology, Department of Molecular Virology (DBM), Université Libre de Bruxelles (ULB), 6041 Gosselies, Belgium; Roxane.Verdikt@ulb.be (R.V.); Olivier.Hernalsteens@ulb.be (O.H.)

**Keywords:** HIV-1 latency, HIV-1 persistence, reservoirs, epigenetics

## Abstract

Eradicating HIV-1 in infected individuals will not be possible without addressing the persistence of the virus in its multiple reservoirs. In this context, the molecular characterization of HIV-1 persistence is key for the development of rationalized therapeutic interventions. HIV-1 gene expression relies on the redundant and cooperative recruitment of cellular epigenetic machineries to *cis*-regulatory proviral regions. Furthermore, the complex repertoire of HIV-1 repression mechanisms varies depending on the nature of the viral reservoir, although, so far, few studies have addressed the specific regulatory mechanisms of HIV-1 persistence in other reservoirs than the well-studied latently infected CD4^+^ T cells. Here, we present an exhaustive and updated picture of the heterochromatinization of the HIV-1 promoter in its different reservoirs. We highlight the complexity, heterogeneity and dynamics of the epigenetic mechanisms of HIV-1 persistence, while discussing the importance of further understanding HIV-1 gene regulation for the rational design of novel HIV-1 cure strategies.

## 1. Introduction

Since its introduction in the mid-1990s, combination antiretroviral therapy (cART) has proven to be a tremendous success in the clinical management of HIV-1 infections. Not only is the therapy suppressive of HIV-1 replication, but it also prolongs the lifespan of infected individuals and has globally decreased AIDS-related morbidities, transforming HIV-1 infection from a deadly to a chronic disease [1,2]. In an ageing population of people living with HIV-1, the current challenge is to eradicate the virus, or, more realistically, to manage a functional cure, thereby limiting the development of HIV-1 co-morbidities and improving the quality of life of HIV^+^ individuals. Indeed, despite cART, HIV-1 persists in treated individuals under several sources collectively referred to as “viral reservoirs”. These reservoirs correspond to cell types or tissue compartments where more stable viral kinetics are maintained in comparison to the pool of cells containing actively replicating viruses [3,4]. Due to their low replication level or their anatomical distribution, viral reservoirs are less sensitive or insensitive to cART drug regimens [5]. However, upon cART cessation or interruption, viral reservoirs can rekindle the infection and will feed the rebounding viraemia [6,7,8]. The existence of HIV-1 reservoirs has thus considerably thwarted attempts to eradicate the virus in HIV^+^ individuals and they are today considered as a major barrier to achieving an HIV-1 cure [4,9].

Latently infected resting memory CD4^+^ T cells were the first identified sources of HIV-1 persistence and are still the best-characterized cellular reservoirs [10,11,12,13]. Latency is a non-productive form of infection wherein HIV-1 gene expression is maintained in a reversible silent state [14]. Several subsets of CD4^+^ T cells, including memory and naïve phenotypes, can be latently infected [15,16,17,18,19]. In recent years, the definition of viral reservoirs has been extended to account for the intrinsic heterogeneity of viral reservoirs depending on the nature of the infected cell, its tissue localization and the state of the HIV-1 provirus (i.e., whether it is competent or defective for viral replication) [20]. Indeed, in addition to the highly heterogeneous CD4^+^ T-cell reservoir, cells of myeloid lineages constitute an underappreciated cellular reservoir of HIV-1 persistence. Myeloid cells are more resistant to HIV-1-induced cytopathic effects and can harbor and release intact virions for a long period of time [21]. However, viral persistence in myeloid lineages, particularly in long-lived tissue-resident macrophages, is probably not accompanied by HIV-1 latency as it is defined in resting memory CD4+ T cells [5]. Indeed, considering the high heterogeneity of HIV-1 reservoirs, molecular mechanisms are likely to differ among the sources of viral persistence. In the present review, we focus on the epigenetic regulation of HIV-1 persistence and we describe it in two types of cellular reservoirs: the latent reservoir of CD4^+^ T cells and the reservoirs of myeloid cells, which we refer as to “persistent reservoirs” (Table 1).

Epigenetic processes are key elements in the silencing of HIV-1 gene expression. Indeed, as with all cellular genes, HIV-1 DNA is hierarchically organized into chromatin. The fundamental structural and functional repeating unit of chromatin is the nucleosome, in which 146bp of DNA are wrapped in superhelical turns around an octamer composed of two copies of each histone protein H2A, H2B, H3 and H4 [27]. Each nucleosome core is linked to the next by a segment of linker DNA that varies in length, from 10 to 80 bp [28]. This nucleosomal array further assembles into higher-order condensed structures, which are stabilized by the linker histone H1 [28]. The state of condensation of the chromatin fiber dictates its functional accessibility to protein machineries responsible for DNA-templated processes, such as transcription [29]. Euchromatin corresponds to an open, less-compacted and accessible state, whereas heterochromatin corresponds to a highly compacted and inaccessible state [30]. The dynamic transitions between heterochromatin and euchromatin thus play a crucial role in gene expression [30]. In this context, the formation of chromatin blocks during latent infections is believed to be one of the primary events leading to transcriptional silencing of HIV-1 gene expression [31]. Epigenetic mechanisms, referring to reversible and heritable changes in gene expression that are due to changes in the chromatin structure without changes in the nucleotide sequence [32], consist of several interrelated processes that cooperatively establish and maintain transcriptional competence [33]. These processes include positioning and remodeling of nucleosomes along the genome, histone modifications, DNA modifications, non-coding RNA (ncRNA)-mediated modifications of the chromatin structure and the three-dimensional organization of chromatin domains in the nucleus.

In this review, we provide an exhaustive and updated picture of the current understanding of epigenetic silencing of HIV-1 gene expression in its different viral reservoirs. We particularly highlight how the virus relies on cellular machineries to cooperatively and redundantly establish and maintain a heterochromatic environment on its promoter. The systematic silencing of HIV-1 gene expression points towards the importance of the viral persistence phenomenon in the virus life cycle. As a consequence, HIV-1 persistence is the target of several anti-AIDS therapeutic strategies. One widely proposed approach, referred to as the “shock and kill” strategy, relies on the use of latency-reversing agents (LRAs) that reactivate HIV-1 gene expression from its latent state [34,35]. Transcriptionally repressed proviruses from the latent reservoirs are reactivated by LRAs in the hope that the infected cells will then die as a result of viral cytopathic effects and/or of HIV-specific immune responses [5]. Over the years, the development of new classes of LRAs has been guided by the precise molecular understanding of HIV-1 silencing mechanisms and several classes of LRAs have been identified, including epigenetic LRAs. In the present review, we will describe some examples of how the epigenetic characterization of HIV-1 persistence mechanisms has led to the evaluation of epigenetic drugs in reactivation assays of HIV-1 gene expression.

## 2. Epigenetic Mechanisms of HIV-1 Latency

During HIV-1 infection and after retrotranscription of the viral RNA genome, the vast majority of viral double-stranded DNA remains unintegrated, mainly under the form of linear DNAs [6,36,37]. These linear HIV-1 DNA molecules serve as precursors for the integration in the host genome but are also susceptible to multiple other fates (Figure 1) [38]. Based on the integration event in the host genome, two forms of HIV-1 latency can be distinguished: pre-integration latency and post-integration latency.

### 2.1. Mechanisms of Pre-Integration Latency

Pre-integration latency refers to silencing mechanisms occurring on 1-LTR circles—which are formed due to homologous recombination of linear HIV-1 DNA at the identical long terminal repeats (LTRs)—or on 2-LTR circles—which are formed by the intervention of cellular DNA repair mechanisms (Figure 1) [38]. Several studies have reported that transcription can occur from HIV-1 1-LTR circles and more extensively from HIV-1 2-LTR circles [38,39,40]. However, upon sensing exogenous viral DNA, invaded cells have developed numerous protection mechanisms, including the induction of an epigenetic silencing onto the foreign DNA [41]. Accordingly, a pioneer study has shown that HIV-1 unintegrated circles adopt an episomal structure, indicating that HIV-1 unintegrated DNA is associated with nucleosomes [42]. More specifically, HIV-1 episomes have been found to be enriched in histone modifications typical of a repressive heterochromatin architecture, such as the trimethylation of histone 3 lysine 9 (H3K9me3) [42]. A recent report has confirmed this observation and shown that histones are loaded soon after the synthesis of HIV-1 DNA and before its integration in the host chromatin [43]. In particular, nucleosome positioning was found to dynamically vary at the 5′LTR between unintegrated and integrated forms of HIV-1 DNA [43]. The more closed nucleosome lattice in the unintegrated viral promoter, in turn, prevented the transcription of viral genes [43]. However, the mechanisms by which epigenetic silencing is exerted on unintegrated HIV-1 episomes remains to be elucidated. In this regard, the cellular HUSH (Human Silencing Hub) complex, composed of the three subunits TASOR, MPP8 and periphilin, has been shown to participate in the epigenetic silencing of unintegrated forms of the murine leukemia virus (MLV) [44]. Through the chromodomain of MPP8, the HUSH complex preferentially localizes to regions rich in the repressive histone mark H3K9me3, where it recruits the histone methyltransferase SETDB1 (SET domain, bifurcated 1) that further catalyzes the accumulation of H3K9me3 [45]. Coincidentally, HUSH has also been shown to participate in the epigenetic silencing of integrated HIV-1 proviruses [46,47] (see below). Hence, an interesting hypothesis would be that HUSH could mediate the transcriptional silencing of both unintegrated and integrated HIV-1 DNA.

Despite the still unclear mechanisms of unintegrated HIV-1 silencing, unintegrated episomal circles can be detected in cART-treated individuals, especially in the context of antiretroviral drugs that inhibit the integration step of the viral replication cycle [37,48]. However, the exact contribution of pre-integration latency to the long-term persistence of HIV-1 in infected individuals is highly debated. Indeed, in contrast with other viral episomes, such as herpesvirus episomes, HIV-1 unintegrated circular cDNA forms lack an origin of replication and are thus lost upon cell division [38]. Therefore, unintegrated HIV-1 DNA is more likely to persist in slow non-dividing cells in vivo, such as certain subsets of memory CD4^+^ T cells [49].

### 2.2. Mechanisms of Post-Integration Latency

Post-integration latency results from multiple interrelated processes that collectively establish and maintain the HIV-1 promoter silence, in a continuum of epigenetic, transcriptional and post-transcriptional mechanisms [31,50,51]. In contrast with the lack of reports on the epigenetic regulation of pre-integration latency, post-integration latency has been extensively studied. In particular, HIV-1 epigenetic silencing during post-integration latency results from: (i) the repressive nucleosome array on the HIV-1 promoter located in the 5′LTR, (ii) the accumulation of histone repressive marks, (iii) DNA methylation of two CpG islands surrounding the transcription start site (TSS), (iv) RNA-mediated mechanisms and (v) the nuclear localization of the provirus in specific chromatin domains.

#### 2.2.1. Nucleosome Positioning on the HIV-1 Provirus

Nucleosomes are dynamic entities and their positioning along the genome directly affects gene expression [52]. Nucleosome positioning is determined by the combined effects of multiple factors, including the DNA sequence itself, DNA-binding proteins and chromatin remodelers [53]. Chromatin remodelers use the free energy available from adenosine triphosphate (ATP) hydrolysis to weaken DNA:histone contacts, resulting in the sliding, spacing, eviction or transfer of nucleosomes from specific regions and the determination of the position and the size of nucleosome-depleted regions (NDRs) along the genome [53]. Based on subunit composition and biochemical activity, five classes of chromatin remodelers are currently distinguished: the BAF/PBAF (BRG1- or HBRM-associated factors/Polybromo-associated BAF), the ISWI (Imitation SWItch), the NURD/Mi-2/CHD (Nucleosome Remodeling Deacetylase/Mi-2/Chromodomain Helicase DNA-binding), the INO80 (INO80 Complex ATPase) and the SWR1 (SWI2/SNF2-Related 1 Chromatin Remodeling) families, which do not all participate in the regulation of gene expression but have also been associated with other DNA-templated processes, such as DNA replication [54].

The BAF/PBAF family of chromatin remodelers generally disorders nucleosomes, thus playing a role in the determination of NDRs [54]. The identification of NDRs has been experimentally guided by the mapping of DNase I hypersensitive sites (DHSs) [55]. Because NDRs are accessible regions that are typically found at transcriptional *cis*-regulatory elements, the mapping of DHSs is reflective of transcriptional competence [56]. Early studies of DHS mapping along the HIV-1 genome indicated that nucleosomes are strictly deposited at specific positions in the latent HIV-1 proviruses, independently of their integration sites in the host chromatin [57,58]. In particular, in the repressed 5′LTR, the enhancer and the core promoter of HIV-1 are, respectively, marked by DHS_2_ and DHS_3_ and surrounded by the two nucleosomes nuc-0 and nuc-1 (Figure 2) [57]. The BAF complex is recruited to the 5′LTR by the short isoform of the bromodomain-containing protein 4 (BRD4S, Figure 2) [59] and is responsible for the active deposition of nuc-1 immediately downstream of the HIV-1 TSS, in an otherwise refractory sequence to nucleosomes [58,60]. At this strategic position, nuc-1 is highly repressive for HIV-1 transcription, and for efficient and productive transcription to occur, nuc-1 must be disrupted [61,62]. In fact, because of its role in maintaining HIV-1 latency by depositing the repressive nuc-1, BAF has been proposed as a target for therapeutical intervention aiming at reactivating HIV-1 gene expression from latency [63]. Interestingly, while very similar in terms of subunits, the BAF and PBAF complexes have distinct opposite roles in the regulation of HIV-1 gene expression. Indeed, whereas BAF restricts HIV-1 transcription, PBAF is required for displacing nuc-1 upon transcriptional activation [58]. PBAF has been shown to be recruited to the 5′LTR by the virally encoded Tat transactivator [64,65] as well as by a Tat-independent mechanism [66]. Finally, the process of BAF/PBAF regulation of HIV-1 gene expression appears to be intimately linked with the process of integration of HIV-1 in the host chromatin. Indeed, the INI-1 subunit (Integrase Interactor 1, also known as SMARCB1, hSNF5 or BAF47) of the BAF complex is known to interact with the HIV-1 integrase [67]. HIV-1 proviral integration and the deposition of repressive nucleosomes could thus be functionally coupled and occurring early in the dynamics of epigenetic latency establishment. In this regard, the histone chaperone Spt6 was also shown to interact with the HIV-1 integration machinery and to participate in the 5′LTR epigenetic repression during latency [68,69,70]. Histone chaperones are histone-binding proteins that are critical in the regulation of nucleosome dynamics, notably by interacting with chromatin remodelers [71]. Spt6 could thus play a pivotal role in coordinating nucleosome occupancy, possibly through interacting with BAF/PBAF, although this still needs to be demonstrated for the latent HIV-1 5′LTR [70].

Of the other chromatin remodeler families known in mammalian cells, so far, only members of the NURD/Mi-2/CHD family have also been linked to HIV-1 latency. Both CHD1 and CHD2 were shown to be important for the regulation of HIV-1 transcription, although the epigenetic mechanisms at play remain elusive [68,72]. Similarly to SWI/SNF-mediated remodeling, specific histone chaperones complexes might be involved in finely tuning nucleosome occupancy and HIV-1 gene expression, as both the HIRA (Histone Cell Cycle Regulator) and the FACT (Facilitates Chromatin Transcription) complexes were also shown to be involved in HIV-1 latency regulation [68].

Collectively, the understanding of the mechanisms governing nucleosome deposition on the HIV-1 promoter during latency, and their possible interplay with other epigenetic marks, as well as with other steps of the virus life cycle, is still at its early beginning. Future studies will need to address how dynamic changes occur in the HIV-1 nucleosomal array during latency establishment, maintenance and reversal. As nucleosome deposition appears to be an early event in HIV-1 silencing, targeting this step might prevent latency establishment in the 5′LTR, hence limiting the formation of viral reservoirs.

#### 2.2.2. Repressive Histone Marks on the HIV-1 Promoter during Latency

In addition to changes in nucleosome positioning, dynamic changes in nucleosome composition also occur during gene expression regulation. Indeed, the basic amino-terminal tails of histones are disordered and protrude from the nucleosome core [73]. Individual or multiple additions or removals of post-translational modifications (PTMs) on histone tail residues are associated with variations in the chromatin structure [73]. Acetylation, methylation, phosphorylation, ubiquitination or sumoylation are some of these histone reversible covalent modifications, catalyzed by separate classes of enzymes [74]. Histone PTMs can directly affect the compaction degree of chromatin. For instance, acetylation of lysine residues by histone acetyltransferases (HATs) neutralizes their positive charge, weakening DNA:histone electrostatic interactions and resulting in an accessible euchromatin structure favorable for transcription [73]. Furthermore, histone PTMs can also indirectly regulate the process of transcription by serving as a scaffold for the binding of effector proteins (that possess bromo-, chromo- or PHD domains to recognize the PTMs) [74]. Alternatively, histone PTMs can prevent the binding of proteins to the chromatin fiber [74]. Thus, alone or in combination, histone PTMs constitute an important mechanism of gene expression regulation, which led to the notion of a “histone code” in the early 2000s [75].

Histone deacetylases (HDACs) are a group of enzymes that erase the acetylation of lysine ε-amino groups [76]. Since histone acetylation is generally correlated with gene activation, histone deacetylation is associated with gene repression, although this also occurs through the deacetylation of other substrates than histones [76]. Based on their enzymatic activity and their cellular sub-localization, human HDACs are classified into four classes (Table 2): the class I HDACs (HDAC1, HDAC2, HDAC3 and HDAC8), the class IIa HDACs (HDAC4, HDAC5 and HDAC7) and the essentially cytosolic class IIb HDACs (HDAC6 and HDAC10), the class III HDACs (or sirtuins) and the class IV HDACs (containing the lone HDAC11). Class I, II and IV HDACs possess a Zn2^+^-dependent enzymatic activity, whereas sirtuins’ activity depends on NAD^+^ [76]. Multiple repressive cellular transcription factors binding to the HIV-1 5′LTR region are responsible for the indirect and redundant recruitment of class-I HDACs during viral latency (Figure 2) [77,78,79,80,81]. For instance, the repressive homodimer of NF-κB p50-p50 binds its cognate sites in the 5′LTR enhancer, where it mediates the recruitment of HDAC1 [78]. Similarly, the Yin Yang 1 (YY1) factor and the late SV40 factor (LSF) also bind the 5′LTR and recruit HDAC1 [77,82], whereas the transcription factor C-promoter binding factor-1 (CBF-1) binds the viral promoter, where it recruits corepressor complexes containing HDAC1 and HDAC3 [81]. In addition, HDAC4 (a class II HDAC) has been shown to repress HIV-1 gene expression in response to environmental stressors, such as the lack of essential amino acids in the environment [83]. However, due to the low deacetylase activity of class IIa HDACs towards histones, this repression could be based on other epigenetic mechanisms than histones PTMs, such as changes in the nuclear organization [83]. Finally, sirtuins have been shown to participate in HIV-1 gene regulation but through other mechanisms than histone deacetylation. In particular, SIRT1 plays an important role in controlling the recycling and the transactivation feedback of Tat [84]. Together, the redundant recruitment of HDACs to the HIV-1 5′LTR during latency explains the success of using broad-spectrum HDAC inhibitors (HDACi) in ex vivo latency reversal strategies [34]. However, with the development of selective and more potent HDACi that have specific activities on certain classes of HDACs [85], a more global picture of which HDACs participate in viral latency will be needed to reverse HIV-1 latency in more targeted approaches. This is particularly important in light of a recent study showing that histone acetylation also surprisingly contributes to HIV-1 latency (Table 2). Indeed, the lysine acetyltransferase 5 (KAT5, also known as Tip60) was found to promote HIV-1 latency through the acetylation of H4 on the 5′LTR, this, in turn, allowing the recruitment of BRD4 and a block in HIV-1 transcriptional elongation (Figure 2) [86]. This latter study illustrates that the hypoacetylation of H3 and the hyperacetylation of H4 have an opposite compatible role in the heterochromatinization of the 5′LTR during latency. Based on these mechanistic insights, the rational development of HIV-1 latency reversal strategies should aim at modulating histone acetylation patterns on the viral promoter by favoring the inhibition of HDACs targeting H3 and of HAT(s) targeting H4.

Methylation of histone tails consists of the addition of one, two or three methyl groups, either on lysine residues of histone tails (H3K4, H3K9, H3K27, H3K36, H3K79 and H4K20) by histone lysine methyltransferases (HKMTs) or on arginine residues of histone tails (H3R2, H3R8, H3R17, H326 and H4R3) by protein arginine methyltransferases (PRMTs, Table 2) [95]. As for HDACs, histone methyltransferases (HMTs) have also been reported to methylate many basic residues in other proteins than histones, thereby participating in the regulation of gene expression [95]. Histone methylation role in transcription depends on the location into specific *cis*-regulatory regions, on the residue targeted and on the number of methyl groups added [95]. In addition, viruses may exploit non-canonical histone methylation pathways, and the methylation of some residues, which is generally associated with transcriptional activation in cellular genes, may not be similarly active in the case of viral gene expression. Regarding histone lysine methylation, both H3K27 and H3K9 methylation have been involved in HIV-1 silencing during latency. The HMT EZH2, which is part of the Polycomb Repressive Complex 2 (PRC2), together with the corresponding H3K27me3 that this enzyme catalyzes, have been reported on the viral promoter in both cell lines and primary cell models for HIV-1 latency (Figure 2) [87]. In addition, both EHMT1 (Euchromatin histone methyltransferase)/GLP and EHMT2/G9a participate in HIV-1 latency by depositing H3K9me2 on the HIV-1 promoter in latently infected T cell lines (Figure 2) [88,89,90]. Together, these studies indicate that H3K27 and H3K9 methylation co-occurs on the latent HIV-1 promoter, suggesting that functional crosstalk between these two epigenetic pathways might be involved to cooperatively establish a heterochromatic environment on the latent 5′LTR. Indeed, while H3K27 and H3K9 methylation is generally considered mutually exclusive and defining facultative and constitutive heterochromatin, respectively, crosstalk between these two epigenetic marks has been reported in the literature [96]. In this case, cooperative recruitment of H3K27 and H3K9 HMTs could then occur [96], which in the context of HIV-1 latency might explain the recruitment of EZH2 and EHMT1/2 to the 5′LTR. A recent report has identified the transcription factor CBF-1 as responsible for the concurrent recruitment of not only these HMTs but also HDAC1 and HDAC3, further supporting the notion that histone methylation and deacetylation are coordinated in HIV-1 silencing [97]. In addition to this, as discussed above, the tripartite HUSH complex also contributes to the spreading of H3K9me3 through the chromodomain-containing MPP8 that “reads” the epigenetic code and recruits the “writer” SETDB1 (Figure 2) [46,47]. A similar mechanism of “heterochromatic spreading” on the latent HIV-1 promoter occurs through the chromodomain-containing HP1γ protein that recognizes H3K9me3 and recruits the SUV39H1 HMT (Figure 2) [91]. More recently, it has also been shown that the known restriction factor APOBEC3A (Apolipoprotein B mRNA editing enzyme catalytic subunit 3A) acts epigenetically in restricting HIV-1 gene expression, independently of its cytidine deaminase activity [98]. Indeed, APOBEC3A was found to bind the NF-κB sites in the 5′LTR, thus promoting the recruitment of KAP1/TRIM28 and HP1 to the viral promoter [98]. This latter report thus provides an alternative mode of recruitment of H3K9 HMTs to the viral promoter during latency. Regarding other histone lysine residues, SMYD2 has been involved in HIV-1 latency by mediating the mono-methylation of H4K20 on the HIV-1 5′LTR, which could then potentially lead to the recruitment of the Polycomb Repressive Complex 1 (PRC1) and further chromatin compaction (Figure 2) [92]. Finally, a recent study has shown that histone demethylases are also involved in HIV-1 latency. Indeed, the histone demethylase MINA53 has been identified in a CRISPR/Cas9 screen as a new actor in HIV-1 latency, through the demethylation of H3K36me3 on the 5′LTR and crosstalk with histone acetylation (Table 2, Figure 2) [94]. Thus, similarly to observations regarding histone acetylation, seemingly opposite histone methylation marks coordinately regulate the heterochromatinization of the latent HIV-1 promoter, which has further implications in the rational design of HIV-1 latency reversal approaches. This is particularly true since the repertoire of histone methylation and possible crosstalk is even more complex because other residues than lysines might be methylated. Thus far, only one study reported the involvement of histone arginine methylation in the HIV-1 promoter silencing during HIV-1 latency. CARM1/PRMT4 catalyzes H3R26 methylation, which has been shown to be associated with the silencing of HIV-1 transcription (Table 2, Figure 2) [93].

Finally, in addition to the well-studied acetylation and methylation, the histone code of HIV-1 latency is more complex than previously thought. Indeed, a recent study has shown that histones in the 5′LTR are hypocrotonylated in latently infected CD4^+^ T cells [99]. Histone crotonylation is a newly discovered histone PTM that consists of the addition of a crotonyl group onto lysine ε-amino groups (Kcr), using crotonyl-CoA, an important intermediate in metabolic pathways, as a cofactor [74]. Accordingly, the hypocrotonylation of the HIV-1 5′LTR in latently infected cells was correlated with lower expression of ACSS2 (Acyl-coenzyme A synthetase short-chain family member 2), an enzyme that participates in the synthesis of crotonyl-CoA [99]. While it remains unclear whether hypocrotonylation of the 5′LTR is actively maintained during latency, aberrant fatty acid metabolism has been linked in HIV^+^ individuals with low ACSS2 expression, potentially favoring the establishment of HIV-1 latency [99].

Together, these studies on histone PTMs collectively reveal that a dynamic and comprehensive picture of the histone code of HIV-1 latency is currently lacking. For instance, whether histone ubiquitination occurs on the 5′LTR during HIV-1 latency and its potential role in viral gene expression still needs to be addressed, especially since components of the ubiquitin–proteasome system have been recently shown to participate in HIV-1 latency [100,101]. Future research will also need to focus on the crosstalk between different histone epimarks since a single amino acid residue can be targeted by enzymes with opposite outcomes on HIV-1 gene regulation.

#### 2.2.3. Implication of DNA Methylation in HIV-1 Latency

DNA methylation is an epigenetic mark that consists, in mammals, in the addition of a methyl group on the fifth carbon of the pyrimidic ring of cytosine residues (5mC), located mainly in the context of CpG dinucleotides [102,103]. When occurring in promoter regions, in dense patches of CpG dinucleotides termed CpG islands (CGIs), DNA methylation is repressive for transcription either directly, by the recruitment of 5mC-recognizing transcription factors, or indirectly, by preventing the binding of positively acting transcription factors [104]. DNA methylation has been involved in a variety of homeostatic biological processes, such as cellular differentiation, imprinting, embryogenesis or inactivation of chromosome X in mammals [103,105]. Furthermore, aberrant DNA methylation profiles are the hallmarks of many diseases, including cancers [106]. In retroviruses, DNA methylation-mediated repression of viral promoters has been involved in the latency of bovine leukemia virus (BLV) and human T cell leukemia virus type 1 (HTLV-1) as well as in the silencing of human endogenous retroviruses (HERVs) [107,108,109]. The HIV-1 promoter region contains two CGIs that surround the TSS, termed the 5′LTR CGI and the non-coding region (NCR) CGI, respectively [110]. Multiple studies since the 1990s have shown that HIV-1 gene repression is associated with DNA hypermethylation of these two CGIs in in vitro T-cell models and in primary cell models for HIV-1 latency (Figure 2) [110,111,112,113,114].

Contrarily to other epigenetic marks, which ex vivo studies are technically challenging, the DNA methylation profile of the 5′LTR has been abundantly studied in cells from HIV^+^ individuals and has been found to be variable [113,114,115,116,117]. The heterogeneous profiles of HIV-1 promoter methylation reported in these studies partly pertain to experimental variables such as the immune typing of isolated CD4^+^ T-cell populations [113,115] or the use of peripheral blood mononuclear cells (PBMCs) [117], which contain more heterogeneous cell populations. Additionally, one study has shown that the accumulation of DNA methylation was especially low in the promoter of replication-defective proviruses, suggesting the existence of differential epigenetic mechanisms of HIV-1 repression in replication-competent versus replication-defective reservoirs [116]. Interestingly, clinical characteristics also appear to influence HIV-1 promoter methylation observed ex vivo. Indeed, hypermethylation of the 5′LTR is positively associated with temporal characteristics, such as the duration of the infection [118] or of the antiretroviral treatment [119]. In agreement, a recent study has highlighted that dynamical changes occur in DNA methylation of the HIV-1 promoter in HIV^+^ individuals, with a general increase in 5′LTR methylation over the span of twelve months of cART uptake [120]. However, this latter study had not addressed the contribution of specific classes of antiretroviral compounds to the 5′LTR methylation accumulation. In addition, HIV^+^ individuals could be subdivided into two groups: one maintaining a high and constant level of DNA methylation on the HIV-1 promoter during the studied period and another group presenting a significant increase in 5′LTR methylation at some point along the studied period [120]. Individuals from the latter group were younger and appeared to better respond to cART in terms of immune parameters (in particular, with higher CD4^+^ T cell/CD8^+^ T cell ratios) compared to the group presenting a constant and high level of 5′LTR methylation) [120]. Collectively, these reports support the notion that temporal parameters affect DNA methylation accumulation on the HIV-1 promoter. These observations are reminiscent of the fact that, during development and tumorigenesis, DNA methylation appears to maintain gene silencing in the long term rather than to initiate the heterochromatic silencing [121,122]. Thus, DNA methylation could play a role in the maintenance of HIV-1 latency rather than the establishment of viral gene repression in the cascade of epigenetic events leading to the 5′LTR heterochromatinization. However, the mechanisms at play, favoring the accumulation of 5′LTR methylation over time, still need to be identified.

Thus far, mechanistic studies have shown that, in latently infected T-cell line models, DNA hypermethylation of the HIV-1 NCR CGI provokes the recruitment of the methyl-binding protein MBD2, which is contained within the chromatin remodeling complex NuRD, also containing HDAC2 [114]. Hence, crosstalk between DNA methylation, nucleosome positioning and histone hypoacetylation participate in the repression of the 5′LTR in latently infected cells. However, how DNA methylation is established on the latent 5′LTR remains unclear. In mammals, three DNA methyltransferases (DNMTs) are responsible for cytosine methylation and are classically subdivided into “de novo” (DNMT3a and DNMT3b) and “maintenance” (DNMT1) methyltransferases [123]. In agreement with this classification, DNA methylation patterns are established in early development by the de novo methyltransferases DNMT3a and DNMT3b and then copied to somatic cells by the maintenance methyltransferase DNMT1 during DNA replication and repair [102]. This model is sustained by DNMT1 preferential affinity for hemimethylated DNA substrates [124] and its regulation along the cell cycle [125] and by the high expression of DNMT3a and DNMT3b in undifferentiated embryonic cells [126]. However, this functional segregation between DNMTs appears somehow oversimplified [102]. Indeed, several studies have shown that DNMTs collaborate and collectively contribute to the dynamic establishment and maintenance of DNA methylation patterns in cellular genes [123]. This might also be the case for HIV-1 since the depletion of DNMT1 has been shown not to be sufficient to completely reduce the level of DNA methylation on the HIV-1 promoter in a T-cell line model for HIV-1 latency [119].

Collectively, accounting for patient-specific variations, DNA methylation is involved in the epigenetic repression of HIV-1 gene expression. However, how DNMTs catalyzing 5mC accumulation on the 5′LTR are recruited still needs to be deciphered. Similar to other epigenetic writers such as HMTs, epigenetic crosstalk between DNMTs and other epigenetic actors might be key for the establishment of DNA methylation patterns on the HIV-1 provirus.

#### 2.2.4. ncRNA-Mediated Mechanisms of HIV-1 Epigenetic Silencing

Several classes of ncRNAs have emerged as important actors in the regulation of gene expression, notably by serving as scaffolds for chromatin-modifying complexes [127]. Regulatory ncRNAs are generally classified depending on their size, with transcripts longer than 200nt termed long ncRNAs (lncRNAs) and smaller transcripts including microRNAs (miRNAs) or piwi-interacting RNAs (piRNAs) [128].

In mammals, small ncRNAs such as miRNAs do not usually rely on epigenetic mechanisms to modulate gene expression [127,128]. Accordingly, multiple cellular miRNAs have been correlated with direct or indirect control of HIV-1 gene expression, but not through direct changes in the proviral chromatin architecture [129,130,131]. For instance, five cellular miRNAs (miR-28, miR-125b, miR-150, miR-223 and miR-382) have been found in one study to directly target the 3’ ends of HIV-1 mRNAs, thus contributing to HIV-1 latency by degrading those mRNAs [129]. In another study, two miRNAs, miR-17-5p and miR-20a, encoded from the polycistronic cellular miRNA cluster miR-17/92, have been found to indirectly restrict the levels of acetylated Tat, thus promoting HIV-1 latency [130]. piRNAs form another class of small ncRNAs that were thought to be expressed exclusively in the germline, where they suppress transposons’ activity by promoting their methylation during spermatogenesis [127,128]. Growing evidence now indicates that piRNAs and the PIWIL protein involved in piRNA synthesis are also expressed in somatic cells, where they regulate gene expression in homeostatic processes as well as in tumorigenesis [132]. In this context, PIWIL4 has been recently found to enforce HIV-1 promoter heterochromatinization through the recruitment of SETDB1, HP1 and HDAC4 [133]. PIWIL4 recruitment to the viral promoter was dependent upon its association with piRNAs but further mechanistic studies are needed to refine the contribution of piRNAs and PIWIL proteins to HIV-1 latency.

Similarly to miRNAs, several cellular lncRNAs have been shown to promote HIV-1 latency, although not specifically by epigenetic mechanisms. For instance, the lncRNA *NRON* indirectly restricts HIV-1 gene expression by inducing Tat proteasomal degradation [134] and the lncRNA *NKILA* (NF-κB-interacting long non-coding RNA) represses HIV-1 gene expression and replication by interfering with the NF-κB signaling pathway [135]. Furthermore, recent studies also showed evidence of an epigenetic-based control of HIV-1 latency by cellular lncRNAs. First, the lncRNA *MALAT1* was found to sequester EZH2, which is part of the PRC2 complex, thereby counteracting the epigenetic repression of the HIV-1 5′LTR (Figure 2) [136]. Second, the lncRNA *HEAL* positively regulates HIV-1 gene expression by promoting the recruitment of HATs to the HIV-1 promoter, both in T cells and in microglial cells, indicating common epigenetic pathways between different types of reservoirs, at least for this specific lncRNA. Together, these reports illustrate the importance of the landscape of cellular lncRNAs in the outcome of HIV-1 infections [137,138]. Since HIV-1 infection also induces variations in the lncRNAs transcriptome [139], a genome-wide screen for the role of cellular lncRNAs in the epigenetic and transcriptional regulations of HIV-1 latency is needed.

Finally, HIV-1 also encodes “lncRNA-like” viral RNAs that act on gene expression and that promote epigenetic silencing of the viral promoter. The role of the HIV-1-encoded antisense RNA *ASP-1* in the epigenetic silencing of the 5′LTR was first proposed in a study showing that downregulation of this transcript is associated with decreased recruitment of DNMT3a, HDAC1 and EZH2 to the HIV-1 promoter (Figure 2) [140]. Recently, it was further shown that the *ASP-1* RNA recruits PRC2 to the 5′LTR, thereby provoking H3K27me3, nuc-1 assembly and transcriptional silencing and promoting HIV-1 latency (Figure 2) [141]. The HIV-1 antisense RNA is thus at the crossroads of multiple epigenetic mechanisms acting in concert to repress the 5′LTR during latency. Another study also showed evidence of a second type of HIV-1-derived RNA in the epigenetic repression of the viral promoter during latency. Indeed, in the presence of external stimuli such as exosomes from uninfected cells, the 5′LTR is only partially silent and the cellular machinery is capable of transcribing through the nucleosomes up to the beginning of the *gag* gene, between the nucleosomes 2 and 3 [142]. The resulting TAR-*gag* ncRNA was further found to be associated with PRC2, SIN3A, HDAC1 and the ubiquitin E3-ligase CUL4B (Figure 2) [143]. In mammalian cells, SIN3A serves as a co-repressor scaffold through its interaction with both specific transcription factors and histone deacetylases [144]. The HIV-1 encoded TAR-*gag* ncRNA might thus function as a cellular lncRNA, by serving as an “RNA machine” that promotes epigenetic silencing of the viral genes, but the specific sequence of mechanisms involved needs to be further investigated.

Collectively, studies have shown that cellular ncRNAs are exploited by HIV-1 to keep its genome transcriptionally silent, highlighting the physiological importance of latency in the virus life cycle. Furthermore, HIV-1 encodes at least two of its own “lncRNA-like” viral RNAs to further promote epigenetic repression by bridging several mechanisms.

#### 2.2.5. Nuclear Position of the HIV-1 Provirus

An additional layer of modulation of gene expression that has gained increasing interest in the past few years resides in the impact of the chromatin three-dimensional organization in the nucleus on gene expression [145]. Indeed, depending on its transcriptional competence, chromatin dynamically transits between higher-order organization forms within different subnuclear compartments [145].

HIV-1 integration occurs predominantly in transcriptionally active regions in the nuclear periphery, in proximity to nuclear pores [146,147]. In particular, a recent study has shown that this spatial clustering of HIV-1 proviruses in CD4^+^ T cells is explained by preferential hotspots of integration in genes proximal to super-enhancers, corresponding to enhancer-rich genomic regions located on the outer shell of the nucleus [148]. Alternatively, HIV-1 can also integrate into inner regions of the nucleus, becoming transcriptionally silent in subnuclear compartments such as promyelocytic leukemia (PML) nuclear bodies [149]. Thus, three-dimensional chromatin organization, while an important determinant for HIV-1 integration, has not been clearly linked to the viral transcription control. Notably, further mechanistic studies will need to address whether dynamic changes can occur in HIV-1 proviruses’ nuclear positions and whether these correlate with changes in the proviral chromatin structure and transcriptional state.

Structural nuclear proteins, such as nucleoporins, have received increased attention for their roles in the control of cellular gene expression and chromatin organization [150]. Mechanistically, structural nuclear proteins are thus potential candidates in the regulation of HIV-1 gene expression in the three-dimensional nuclear space. In this regard, a study has recently shown that the inner nuclear membrane protein SUN2, in association with lamins, tethers nuc-1 and nuc-2, thereby maintaining a repressive heterochromatic environment on the 5′LTR [151]. However, how the structural components of the nucleus establish and maintain this heterochromatic environment on the HIV-1 promoter during latency still needs to be assessed in detail.

Collectively, multiple studies have reported that HIV-1 integration favors specific nuclear sub-compartments. However, it still remains to be determined whether the chromatin state of the integration loci affects HIV-1 proviral chromatin architecture. Another open question is whether the HIV-1 provirus transits between different higher-order nuclear sub-compartments depending on its transcriptional state.

## 3. Epigenetic Persistence in Myeloid Lineages

In addition to the well-characterized reservoir of latently infected resting memory CD4^+^ T cells, a heterogeneous pool of cell types participates in vivo in HIV-1 persistence in infected individuals [25]. Cells of myeloid lineages, especially tissue-resident macrophages, constitute a systemic and plastic subset of target cells for HIV-1 infection [152]. HIV-1-infected macrophages are also involved in viral persistence, importantly contributing to tissue reservoirs, including in the nervous, pulmonary, cardiovascular, gut and renal organs [153,154]. For instance, microglial cells, the resident macrophages in the central nervous system, are partly protected from the penetration of antiretroviral drugs due to the blood–brain barrier and constitute a reservoir for a pool of evolving HIV-1 quasispecies [155]. Understanding how HIV-1 gene expression is regulated in these non-classical reservoirs is crucial to obtain a more comprehensive picture of HIV-1 persistence, with the hope of achieving a cure.

Contrarily to the vast number of studies addressing the epigenetic regulation of HIV-1 in latently infected CD4^+^ T cells, little research has been conducted on specific epigenetic mechanisms of HIV-1 persistence in cells of myeloid lineages. Unintegrated HIV-1 episomes persist in macrophages [156]; however, their epigenetic and transcriptional states have not been thoroughly studied. Hence, a remaining open question is whether HIV-1 episomes and proviruses are distinctly regulated in myeloid vs. T-lymphoid reservoirs.

CTIP2 (COUP–TF interacting protein 2)/BCL11B is a cellular transcription cofactor that presents pleiotropic functions in HIV-1 gene repression in microglial cells. First, CTIP2 participates in HIV-1 transcriptional repression in microglial cells by recruiting a multi-enzymatic chromatin-modifying complex that establishes a heterochromatic environment at the HIV-1 promoter, in a Tat-independent manner [157,158] (Figure 3A). Indeed, in these cells, CTIP2 is physically recruited to the HIV-1 promoter by its interaction with Sp1 bound to the GC-boxes of the 5′LTR [159], which is made possible by the histone lysine-specific demethylase 1 (LSD1) also bound to the Sp1 sites [160]. LSD1 recruits the multicomponent COMPASS complex, which ultimately leads to the accumulation of H3K4me3, surprisingly associated with transcriptional repression of the HIV-1 promoter [160]. In parallel, CTIP2 sequentially recruits HDAC1 and HDAC2 that deacetylate H3, then the histone methyltransferase SUV39H1 that promotes H3K9me3 [157]. Similarly to T cells [91], through its chromodomain, HP1γ recognizes H3K9me3 and further recruits SUV39H1 to spread heterochromatin across the HIV-1 promoter region [157]. A second mode of HIV-1 gene repression by CTIP2 in microglial cells is through the sequestration of the positive elongation factor b (P-TEFb) cooperatively with HMGA1 (High Mobility Group AT-hook 1) associated with the 7SK small nuclear RNA and HEXIM1 (Figure 3B) [161,162]. While this repressive mode is not strictu senso an epigenetic mechanism, the regulation of HIV-1 transcription elongation by the cellular ncRNA 7SK could be considered as an RNA-mediated epigenetic mechanism. Finally, the transcription factor HIC1 (Hypermethylated In Cancer 1) interacts physically with CTIP2 and HMGA1 to mediate a third repression mode of the HIV-1 5′LTR in microglial cells (Figure 3C) [163]. Indeed, HIC1 downregulation has been shown to be associated with a Tat-dependent increase in HIV-1 gene expression in microglial cells, arguing in favor of a repressive function of HIC1 [163]. Because SIRT1 is a co-repressor of HIC1 [164] and a co-activator of Tat [31], HIC1 Tat-dependent repression of HIV-1 gene expression could be associated with the HDAC function of SIRT1. Thus far, SIRT1 deacetylation of HIC1 was found to be crucial for its Tat-dependent repression of HIV-1 gene expression. However, in this case, no specific role of SIRT1 in 5′LTR histone status could be shown. Collectively, CTIP2 is implicated in the epigenetic and transcriptional repression of HIV-1 in microglial cells by at least three independent mechanisms. In addition, our laboratory has recently reported that CTIP2 interacts with KAP1/TRIM28 and that these two factors cooperate to repress Tat activity [165]. As KAP1/TRIM28 has been shown to favor HMT recruitment in CD4^+^ T cells [98]; a functional interplay between CTIP2 and KAP1/TRIM28 in the epigenetic control of the 5′LTR in microglial cells is expected but needs to be further studied. In addition, it remains to be determined how CTIP2 can switch from one repressive mode to another and whether CTIP2 mediates similar repression mechanisms in latently infected CD4^+^ T cells.

To date, few studies have addressed the specific epigenetic mechanisms of HIV-1 persistence in myeloid cells, especially due to technical constraints such as the tissue-resident nature of macrophages. However, with increasing evidence of the physiological relevance of myeloid reservoirs for HIV-1 persistence in infected individuals [22], further basic research will be needed to determine the myeloid-specific molecular mechanisms of HIV-1 persistence. This, in turn, should allow the development of targeted anti-HIV therapeutic interventions that will account for the heterogeneity of the viral reservoirs.

## 4. Conclusions

HIV-1 persistence, initially considered as an epiphenomenon, now appears to be an integral part of viral pathogenesis [166]. Clinically, HIV-1 persistence, which prevents viral eradication, is being tackled by concerted scientific efforts [167]. Studies of the mechanisms regulating HIV-1 gene expression in its different reservoirs highlight key properties of viral persistence, providing a molecular rationale to search for an HIV-1 cure.

One property of HIV-1 persistence is the complexity of the molecular mechanisms involved. Several interrelated mechanisms are acting at different levels of the gene regulation flow and, as discussed here for epigenetic mechanisms, are acting in concert to cooperatively repress HIV-1 gene expression. With the growing number of studies identifying novel mechanisms of HIV-1 persistence, the molecular repertoire of viral gene repression needs to be constantly re-evaluated. Indeed, so far, independent studies have shown individual mechanisms of HIV-1 gene repression. However, the next challenge in mechanistic studies on HIV-1 persistence will be to obtain a comprehensive picture of the crosstalk between these mechanisms. This, in turn, should help in the rationalization and the modeling of novel therapeutic interventions. For instance, considering the quantity of redundant recruitments of epigenetic modifiers in the 5′LTR heterochromatinization, any strategy aiming at reversing HIV-1 latency should simultaneously target different mechanisms in combination for synergistic HIV-1 reversal [93,168].

A second key property of HIV-1 persistence is its heterogeneity: not only is the repertoire of molecular mechanisms of HIV-1 persistence complex, but it varies in different cell types and for different infected individuals. This heterogeneity of the HIV-1 reservoirs, a property that substantially hinders the design of a cure, intrinsically results from HIV-1’s adaptation to different cellular environments of infection. Hence, targeted approaches, adapted for each type of reservoir and for each individual, will need to be considered for a cure. In this regard, the vast majority of mechanistic studies on HIV-1 persistence have been performed in CD4^+^ T-cell latent reservoirs, with little understanding of the specific regulation of HIV-1 gene expression in other persistent reservoirs.

There is also increasing evidence towards the importance of a temporal property in HIV-1 persistence: for the same individual, HIV-1 gene regulation will be differentially modulated over time. Hence, the molecular mechanisms of HIV-1 persistence, establishment and maintenance might differ during infection. However, the sequence of events that can dynamically switch the viral promoter between active and inactive states has not been thoroughly addressed so far. With an exciting recent work showing that viral cell-to-cell transmission primes T cells to latent HIV-1 infections [169], another important challenge will be to determine how the repressive epigenetic status of the HIV-1 promoter is established in the early stages of the infection. This will provide further insights into the interplay between the epigenetic control of the virus and epigenetic alterations in the host, which might have fundamental consequences for HIV-1 clinical management [170].

In conclusion, while much knowledge has been garnered on the epigenetic control of HIV-1 latency, mechanistic studies have highlighted the complexity, the heterogeneity and the dynamics of HIV-1 persistence. Further molecular understanding of these properties of HIV-1 persistence will be essential in the search for an HIV-1 cure.

## Figures and Tables

**Figure 1 vaccines-09-00514-f001:**
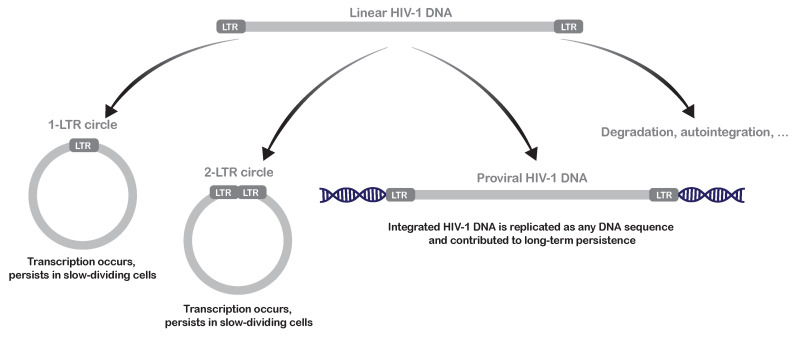
Fates of unintegrated HIV-1 DNA and respective contribution to viral persistence. Adapted from [38]. Following retrotranscription, the nuclear linear HIV-1 DNA can be integrated into the host genome as a provirus, can be degraded or can recombine or exploit host DNA repair machinery to generate functional 1-LTR or 2-LTR episomal circles from which transcription arises.

**Figure 2 vaccines-09-00514-f002:**
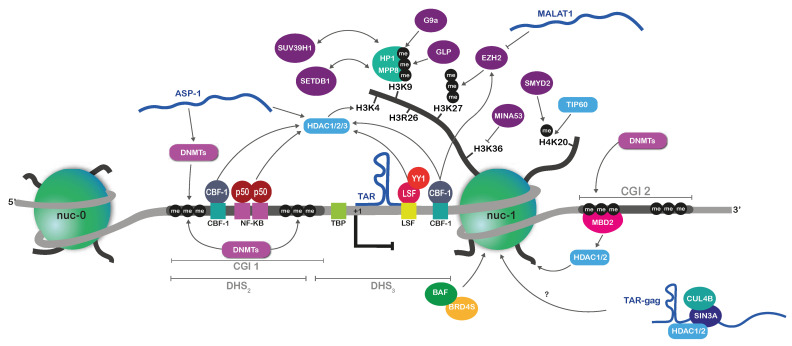
The 5′LTR is heterochromatized in HIV-1 latently infected CD4^+^ T cells. A multitude of interrelated epigenetic mechanisms cooperatively maintain the HIV-1 promoter in a heterochromatic architecture in latently infected CD4^+^ T cells. These include the position of repressive nucleosomes on the 5′LTR, the presence of repressive histone marks such as hypoacetylation or H3K9me3, the hypermethylation of two CpG islands surrounding the transcription start site and the involvement of lncRNA-like epigenetic mechanisms.

**Figure 3 vaccines-09-00514-f003:**
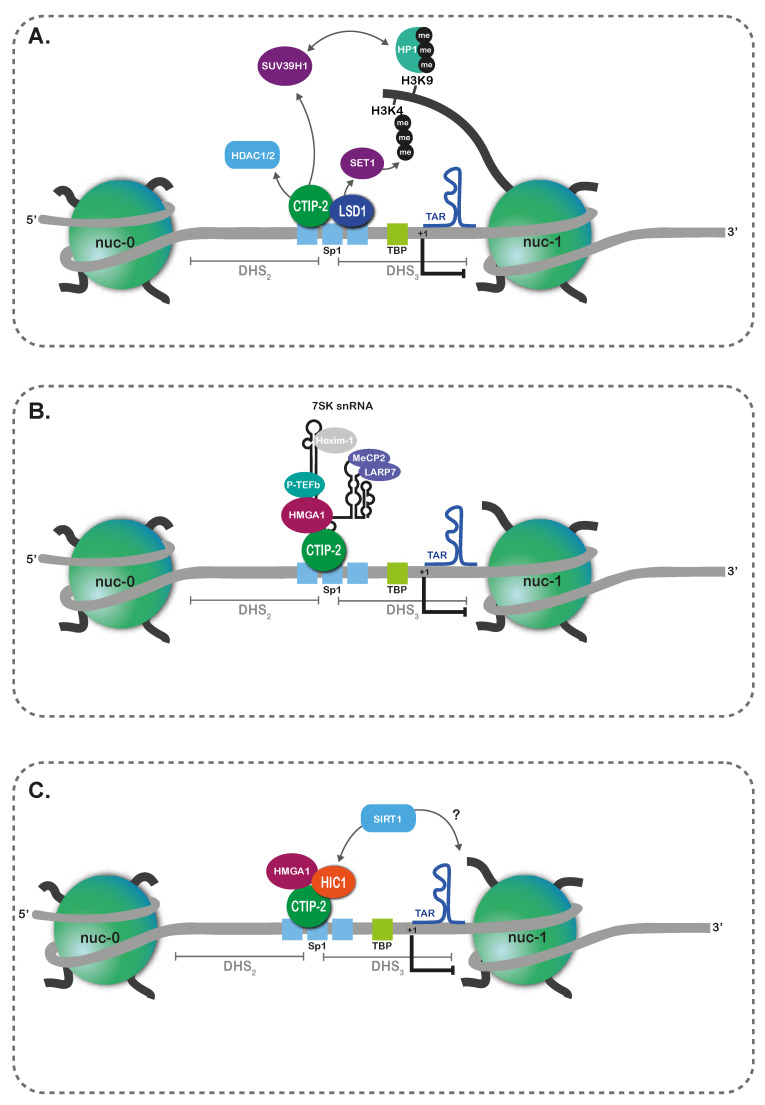
The multiple modes of CTIP2-mediated repression of HIV-1 gene expression in microglial cells. CTIP2 presents pleiotropic functions in HIV-1 gene repression in microglial cells. (**A**) CTIP2 and LSD1 bind the Sp1 sites in the 5′LTR. CTIP2 recruits sequentially HDACs and the HMT SUV39H1 that catalyzes H3K9me3. This mark is recognized by HP1 that recruits more units of SUV39H1 that spread the heterochromatic mark. In parallel, LSD1 recruits the hCOMPASS complex, containing notably the HMT SET1 that stimulates H3K4me3. (**B**) CTIP2 associated with HMGA1 stabilizes the inactive P-TEFb complex (composed of the small nuclear 7SK RNA, HEXIM-1 and LARP7 and MeCP2). (**C**) HIC1, CTIP2 and HMGA1 form a tripartite repressive complex that may be due to SIRT1 HDAC activity.

**Table 1 vaccines-09-00514-t001:** Latent and persistent reservoirs for HIV-1.

Characteristics	Latent Reservoirs	Persistent Reservoirs
Cell type	CD4^+^ T cells (e.g., T_CM_ [6], T_N_ [19], etc.)	Myeloid cells (e.g., macrophages [22,23]), Follicular dendritic cells [24], Epithelial cells? [25] Tissue-resident memory CD4^+^ T cells (T_RM_)? [26]
Causes for cART inefficiency	Low or no replication	Tissue localization and poor drug penetration
Time of establishment	Early during the infection	“Mature” reservoirs
Epigenetic mechanisms of HIV-1 gene regulation	Extensively studied	Poorly studied

**Table 2 vaccines-09-00514-t002:** Histone modifiers implicated in HIV-1 latent CD4^+^ T-cell reservoirs.

	**Histone Deacetylases**	
**Family**	**Members ^†^**	**References**
Class I	**HDAC1, HDAC2, HDAC3**, HDAC8	**[77,78,79,80,81,82]**
Class IIa	**HDAC4**, HDAC5, HDAC7, HDAC9	[83]
Class IIb	HDAC6, HDAC10	
Class III (Sirtuins)	**SIRT1**-7	[84]
Class IV	HDAC11	
	**Histone Acetyltransferases**	
**Family**	**Members ^†^**	**References**
GNAT	KAT2A/GCN5, KAT2B/PCAF	
MYST	**KAT5/TIP60**, KAT6A/MOZ/MYST3, KAT6B/MORF/MYST4, KAT7/HBO1/MYST2, KAT8/MOF/MYST1	**[86]**
p300/CBP	KAT3B/p300, KAT3A/CBP	
	**Histone Methyltransferases**	
**Family**	**Members ^†^**	**References**
HKMTs	ASH1L, DOT1L, **EHMT1-2**, EZH1, **EZH2**, MLL1-4, NSD1-3, SETD1A, **SETD1B**, SETD2, SETD7, **SMYD2**-3, **SUV39H1**-2, SUV420H1-2	[87,88,89,90] [46,47] [91,92]
PRMTs	**CARM1/PRMT4**, PRMT1, PRMT5-7	[93]
	**Histone Demethylases**	
**Family**	**Members ^†^**	**References**
KDM	LSD1/KDM1A, LSD2/KDM1B	
JMJD	KDM2-8 classes that contain over 30 members, including **MINA53/JMJD10**	[94]

^†^ Enzymes indicated in bold and underlined have been linked to HIV-1 silencing during latency, according to the references presented in the column on the right. HKMT, histone lysine methyltransferase; PRMT, protein arginine lysine methyltransferase; KDM, lysine demethylase; JMJD, JumonjiC domain-containing histone demethylase. Of note, only enzymes involved in histone acetylation and methylation associated with the repression of the 5′LTR are listed in the table.

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
