# Peer review of "Epigenetic Mechanisms of HIV-1 Persistence"

_vaccines, 2021, doi:10.3390/vaccines9050514_

Round 1

Reviewer 1 Report

Verdikt et al have compiled an interesting review of epigenetic regulation of HIV-1 latency in their manuscript, “Epigenetic Mechanisms of HIV-1 Persistence.” Over all the material is well organized and appropriate.  I have a few comments and suggestions regarding the content that I would like to see clarified below.

  1. “Combinatory antiretroviral therapy (cART)” should be “combination antiretroviral therapy (cART)”
  2. Abbreviations on lines 148 & 149 should be written out. Also on lines 188, 189, & 192. Perhaps an abbreviation list should be added.
  3. Line 233: “Ying-Yang” should be “Yin-Yang”
  4. Please add a brief description on the HIV latency reversal strategies that are being tested (line 246). I’m familiar with the concept, but many readers may not be familiar.
  5. Lines 366-371: Please add more detail in terms of which ART drugs altered HIV methylation. Is the differential methylation induced by ART drug class-specific?
  6. The sentences starting in lines 371 and continuing through 379 are confusing. What does “a more important increase” mean? What do you mean by CD8 T cell decrease?  Does this refer to absolute numbers of circulating cells or percentages?  Please clarify these sentences.
  7. I’m a little confused by Table 1. For “Long-term persistence” it says “Yes” for latent reservoirs and “No” for persistent reservoirs. Why would they be called persistent reservoirs if they lack long-term persistence? When this term is introduced at around line 50, the authors should elaborate on the nomenclature, especially as it pertains to Table 1.
  8. Figures 1 and 2 need to be higher resolution. Figure 2 is especially blurry.

I do have a few concerns with the writing. I have included a couple of examples below. Overall, the manuscript needs editing for English.

  1. There are many instances where a plural is used, but it should be a singular. Examples: line 283 should be “cross-talk” & line 284 should be “recruitment.” Also, there are many instances of “histones” being plural where a singular is preferred, like on line 288
  2. There are many instances of verb cases not being correct (subject-verb agreement). Examples: Line 286: “identify” should be “identified”; Line 299: “provide” should be “provides”

Reviewer 2 Report

The manuscript by Verdikt, Hernalsteens and Van Lint is an extensive review of epigenetic mechanisms of HIV-1 persistence. The authors discuss nucleosome positioning, histone modifications, DNA methylation, non-coding RNA mediated control of HIV transcription and the role of nuclear organization in HIV latency. The review is overall very clearly written and covers the major mechanisms of HIV transcriptional control. I have some relatively minor suggestions to further improve this review.

  1. It appears that the authors define “latency” in CD4+ T cells as strictly silent provirus, and refer to reservoir in other cell types as “persistent”. What about “persistent” reservoir in CD4+ T cells? Multiple studies have demonstrated that the reservoir in CD4+ T cells is not necessarily silent, including intact and defective proviruses (e.g. PMID 28416661, 27432972). Since these persistent proviral states are likely to be controlled by some mechanisms beyond epigenetics, they would not be in the scope of this review, but their existence should probably be discussed.
  2. Another long non-coding RNA that participates in epigenetic regulation of the HIV LTR is NKILA (PMID 32581100). This study should probably be included in the discussion of HIV transcriptional regulation mediated by lncRNAs.
  3. Please clarify: “In addition, two different temporal methylation patterns could be observed among the studied participants, with some HIV+ individuals having a constant increase while other having a more important increase in 5’LTR methylation over time” (lines 371-373 on page 8). What is compared to what here? Short-term vs long-term methylation? Methylation at the LTR vs elsewhere in the HIV genome? What is meant by “a more important increase”? More significant?
  4. Please clarify: “Collectively, the observation that temporal parameters affect DNA methylation accumulation on the HIV-1 promoter follows reports made during development and tumorigenesis, where DNA methylation appears to lock gene expression on long-term rather than to initiate the heterochromatic silencing” (lines 376-378 page 8). What is meant by “to lock gene expression”?
  5. This sentence is too long and somewhat hard to follow, need to read it several times: “Collectively, multiple studies show that HIV-1 integration favours specific nuclear sub-compartments but open and exciting questions that remain to be addressed are whether the chromatin state of integration affects HIV-1 proviral chromatin architecture and whether HIV-1 proviruses transit between higher-order nuclear sub-compartments depending on its transcriptional state” (lines 502-506 on page 10). It may be useful to revise and break up into shorter sentences.
  6. There are some grammatical errors in the text, e.g. “report thus provide” (line 299 page 6) and word usage, e.g. “similarly than in T cells” (line 537 page 11), “provokes” (line 385 page 8 and lines 455-456 page 9), etc.
  7. “Promoter of HIV latency” (line 306 page 6) – the term “promoter” may be misleading in this context since expression regulation at promoters is the main topic here.
  8. “i.e. higher CD4+ T cell gain, higher CD8+ T cell decrease, …” (line 375 page 8) – please revise “dot dot dot”.
  9. Table 2: the subsections indicate different functional enzymes, but functional class is not indicated (e.g. HDACs, methyltransferases, etc.) Repeating subheadings such as “family”, “members”, “references”, is probably not necessary and it may be useful to spell out the class of enzymes instead.

Reviewer 3 Report

The manuscript is lucid for the reasons such as the easy reading flow from starting to end and graphic representation of the various mechanisms. Many of the review articles that I came across focused on only one of the epigenetic mechanisms regulating HIV-1 latency. In contrast, this manuscript explained in-depth information about different epigenetic pathways. Furthermore, the manuscript summarized the current topics in HIV-1 latency, such as non-coding RNA-mediated epigenetic silencing and HIV-1 persistence in the myeloid population. Finally, this manuscript is collectively a one-stop to acquire current knowledge on the host cell epigenetic mechanisms for HIV-1 dormancy.

Minor comments:

  1. Rephrase the incomplete sentence in lines 110-113 (In this regard, the cellular HUSH… forms of murine leukemia virus").
  2. In line 267, replace the word "into" with "of."
  3. Statement in line 447 is confusing, "since HIV-1 infection also induces variations in the IncRNOme". IncRNome is the name of the database for long non-coding RNAs in humans. Does the author intended to write "since HIV-1 infection also induces variations in the IncRNAs"?
  4. In line 537, replace the word "than" with "to."
  5. Line 563 alternative word choice "few" instead of "little."
  6. Many references have the word "Internet" included in them. Please fix the issue.
  7. According to instructions to authors, references should be placed in square brackets. Change the bracket's style.
  8. Include the borders in table 1 to fix the spillover of text from one row to another. Borders will fix the issue and visually allure the readers.

Round 2

Reviewer 1 Report

Verdikt et al have provided a much improved manuscript. Epigenetics of HIV latency is thoroughly reviewed, and material is well presented.